# Attention based GRU-LSTM for software defect prediction

**Hafiz Shahbaz Munir, Shengbing Ren** [ID]**\*, Mubashar Mustafa, Chaudry Naeem Siddique, Shazib Qayyum**

Computer Science and Engineering, Central South University, Changsha, China

\* rsb@csu.edu.cn

## Abstract

Software defect prediction (SDP) can be used to produce reliable, high-quality software. The current SDP is practiced on program granular components (such as file level, class level, or function level), which cannot accurately predict failures. To solve this problem, we propose a new framework called DP-AGL, which uses attention-based GRU-LSTM for statement-level defect prediction. By using clang to build an abstract syntax tree (AST), we define a set of 32 statement-level metrics. We label each statement, then make a three-dimensional vector and apply it as an automatic learning model, and then use a gated recurrent unit (GRU) with a long short-term memory (LSTM). In addition, the Attention mechanism is used to generate important features and improve accuracy. To verify our experiments, we selected 119,989 C/C++ programs in Code4Bench. The benchmark tests cover various programs and variant sets written by thousands of programmers. As an evaluation standard, compared with the state evaluation method, the recall, precision, accuracy and F1 measurement of our well-trained DP-AGL under normal conditions have increased by 1%, 4%, 5%, and 2% respectively.

**Data Availability Statement:** https://zenodo.org/record/2582968#.X5TCBygzY2w.

**Funding:** This paper has no financial support by the funders. This is a pre-research project.

## Introduction

Software largely depends on where you are and the type of life you live in. Software is an important functional component, man-machine interface, and it is also the most unique and valuable part of the solution. A software defect is an infected part of a system program, which sometimes terminates the program unexpectedly, or may help hackers control your program, which may have a devastating effect on the quality and safety of the software. Nowadays, everyone believes more and more in software programs in every field. As a result, recent software programs have become more complicated and expensive [1]. As the size and complexity of software increase every day, it is difficult to detect defects in software code. The importance and challenges of defect prediction have made it a dynamic research field in software engineering [2]. Software Defect Prediction (SDP) in software engineering is one of the most important research fields. It has aroused the curiosity of many researchers in academia and industry [3].

Machine learning technology is easy to build defect prediction models. Machine learning techniques such as Naive Bayes, Random Forests, and Support Vectors are used to derive

**Competing interests:** This paper has no Competing Interests. Because this is a pre-research project, which has no financial support.

various functions for software feedback and encode them into a general category. A lot of research [4] wisely absorbed design features that can distinguish defective code from non-defective code, such as code size and complexity [5] (e.g., MOOD function, Halstead Features, CK features, McAbe), micro-interactions, [6] code loss metric, such as the total number of lines of code changed), smelly statements, [7] and process metric. However, these functions and technologies cannot replicate the semantics and syntax of the program. In addition, the software's metric function is usually not widely used with well-defined functions, because in some software projects [8], employers may not perform well in other projects. Because the semantic and syntactic information are not similar [9]. Functions that include these types of structural information and the semantics of defect prediction should improve performance. The rich functions of code semantics and syntactic structure have specific statistical functions, and ASTs [10] hides these specific statistical functions, which can help locate and analyze faults more accurately.

The benefits of SDP in the research society are widely recognized; however, serious criticisms of SDP are limited world applications [11]. For this method, a root cause seems to be crucial, and it relies on failure prediction at a high level of granularity. Developers need to detect and locate faults that take unimportant time in modules that have been classified as prone to faults. In order to identify the location of the failure during the acceleration process, fine-grained failure prediction is used [12]. We assume that the precise location of error-prone locations through sentence-level SDP has sufficient potential granularity. Therefore, defects can be detected and located with less influence and time.

In our proposed model, we use statement-level granularity and attention-based GRU-LSTM (DP-AGL) for defect prediction.

1. Our goal is to classify faults and continuously learn to improve the accuracy of deep learning.

2. We first use Clang to parse the source code into AST, and introduce 32 level matrix features and tags for each statement, because the feature is the number of unary operators or operands used in each sentence.

3. As the learning part of DP-AGL, we use gated recurrent unit [13] and long-short-term memory with attention mechanism.

The improved version of the LSTM unit is the "gated recurrent unit" [6]. The input and forget gate of the GRU unit are combined into one gate, called the update gate. GRU also combines internal state with its temporary output. By using the previous hidden state, after the gate control is completed, there are most of the differences in the complete gate control unit. Therefore, the bias adopts various simplified forms and combinations, which are called minimum gate control units. The fully gated unit is parallel to the smallest gated unit, rather than parallel to the reset gate, and the updated vector is merged into the forget gate [14]. Sequential defect prediction also requires memory.

This function of failure prediction provides practical utility due to the less troublesome or flawed data in its collection, and these instructions will prevent companies from using [15] for training in the SDP method.

To verify our experiment, in Code4Bench, we selected 119,989 C/C++ programs [16]. The Code4Bench benchmark contains every version of every program we wrote to introduce metrics. For insufficient data, we label each statement as a morpheme; we use equivalent ready-made statistics by using code4bench. After that, we make a matrix for each row of each program, dedicated to press releases. The equivalent column is used for each statement of the

metric data (including lexemes). The label indicates no failure or prone to failure, and failure data [1]. Finally, we train the data on the novel model.

## Background and related work

A brief explanation about fundamental concepts discussed will be in this segment, which supports the paper for state of the art techniques. Then, we put the central part of the literature review that is software defect prediction. We also wrap up this section with specific gaps and lesson-learned.

### Traditional concepts

The prediction of software defects is a noteworthy research area in software engineering (SE) [17]. SDP can automatically anticipate the parts of the software that are prone to defects for effective software trial [18]. Software defect prediction technology supports software metrics and error data to build predictable [19] models. Incorrect data may arrive from other projects, or the same previous version of the project to training data [20]. The resulting model provides a task that comes from invisible software and is used to predict error-prone parts. For example, a similar software program to be released is using [21]. When we obtain the same training data as the project's predictive model, we may implement a project Within Project Defect Prediction (WPDP). On the contrary, when most or all of the training data is obtained from other similar projects, we will obtain a cross-project defect prediction system (CPDP) [21].

### Literature review

**General review in defect prediction.** Most of the references focused on designing new identification functions, filtering fault data, and building effective classifiers. Ball and Nagappan [22] proposed customer churn indicators and attached them to software-dependent defects and failure predictions. Moser et al. [23] introduced the efficiency of detailed analysis and capability of static code attributes and changed metrics for defect or failure prediction. In addition, Ayan and Arar [24] proposed a naive Bayes technique suitable for selecting suitable features for unnecessary filters. Mousavi et al. [25] solved the problem of class imbalance in software failure prediction through integrated learning. Besides, Jing et al. [26] proposed a process of dictionary learning, which requires calculating the cost of misclassification to predict software defects.

Kamei etc. [12] observed the real-time performance of the model from the perspective of cross-project SDP. In their research, they used 11 open source projects. They also developed predictive models because they are built with finer granularity and can be simplified using process metrics. They also introduced cross-projects, where instant models tend to have better performance in the environment. In contrast, the model is trained on the same project, and the amount of project data used, combined, and grouped is derived from many models and developed on many projects.

Khoshgoftaar et al. [27] defined debugging churning as the number of code blocks or lines changed or added to fix errors. Their motivation was to mark modules where debug code loss exceeds a threshold to be classified as vulnerable to defects. They studied the large-scale telecommunications legacy system that was released twice in a row. The system consists of more than 38,000 methods consisting of 171 modules. Differentiating and analyzing modules that identify easy defects support six static software package metrics. When their model was used in the second edition, the misclassification rates of type II and I were 19.1% and 21.7%, respectively. And the overall misclassification rate was 21.0%.

Turhan et al. [15] introduced neighbor filter technology to get rid of those instances of the source item whose characteristics are not close enough to those of the target item. Researchers can use Transmission Component Analysis (TCA +) [28] to formulate a single goal of prediction technology. However, large amounts of data on unrelated projects usually lead to a reduction in regulation. Most researchers focus on the source project's function or filter instances that are unrelated to the target project to solve this problem. Besides, Yu et al. [29] chose characteristics based on correlation to select those strongly related to the target item.

Canfora et al. [30] used SDP's multi-goal formulation problem. The primary purpose was to improve the recall performance and accuracy of the model. However, in the single objective formula, the objective of a useful model is often insufficient. Developers are responsible for checking the categories that are prone to defects and assigning time and energy. These categories and roles play an important role; the larger class is so beautiful that the hope is more significant. Therefore, the researcher recommended checking the best performance of the model to identify faults, thus saving developers' time. Then, they integrated this information into multiple targets for SDP.

Ma etc. [31] proposed a method of using gravitational data to transmit naive Bayes, on which [32] developed a naive Bayes classifier and standardized the weights of coach examples. Some studies have recently shown that if we use a small part of the labeled data in the target project, it will bring higher predictive performance.

Choudhary et al. [33] researched to study the changes in metrics and the code that supports the precise measurement of the SDP model. Researchers use various versions of Eclipse projects as experimental subjects. Besides, to take advantage of significant changes in metrics, many novel metrics have been proposed. They also observed that the SDP model's new change indicators' performance is better than the leading metrics of the SDP model. Also, an advantage is provided by changing the building metrics using the SDP model of high-performance metrics.

Shippey et al. [34] considered the flawed Java code extraction features of the SDP model to improve accuracy. To understand this, they used a bottom-up approach and applied it to the Abstract Syntax Tree (AST) n-gram. They influenced non-parametric testing to detect the association between AST n-grams and software failures. They also used open-source systems and commercial software for subjective testing.

Finally, Chen et al. [35] first started the source item's data weight with the information of the gravitational method, and then adjust it using a limited number of marker features inside the target item through the building of a predictive model called TrAdaboost [36]. Qiu et al. [37] used the kernel mean matching (KMM) algorithm to construct the unique weight of the multi-component learning model. It was divided into multiple parts of the source project data, and KMM was used to adjust the source instance's weight in each part. After that, it uses a part of the labelled data and the source instance with weight for each component in the target instance to build a predictive model. Finally, it can be optimized and prepared for source component weights to develop more accurate integrated classifiers.

**Deep learning in SDP.** As conventionally fault prediction, the data set that is planned by manual measurement features and manual features may not be used for quality insurance or has no significant association with category labels. These characteristics may be pretentious manifestations of prediction. These metrics cannot obtain syntactic and semantic information from the code.

We can express the syntax and semantic information of the code in two ways. One is the Control Flow Graph (CFG), and the other is the Abstract Syntax Tree (AST) [38]. The AST of the program meticulously shows the high-level relationship between different parts of the code. Pan et al. [39] extracted CFG from the assembly code of the project and designed a

convolutional graph network to study the semantic features of the program. CFG represents the code of the software, which is displayed during program execution, and all paths can be traversed. Dam etc. [2] assembled a software defect prediction algorithm based on AST based on the deep tree. Fan et al. [10] the attention-based recurrent neural network is applied to the vector encoding structure of the code as a deep learning model. They built the AST code and converted it into a digital vector. Wang et al. [18] used DBN to obtain hidden functions, which contain the semantics and syntax of the program and provided preprocessed input for the classifier to predict defective codes. Qiao and Wang [40] to implement instant SDP, which used deep learning technology. They made the deep learning model suitable for the SDP context and were mainly able to select operational input structures when the relationship between input and output was complex.

Pan etc. [41] used the upgraded version of Convolutional Neural Network (CNN) by using the in-project SDP and evaluated 12 different versions of the project. They compared it with the SDP method based on baseline deep learning. Li et al. [42] Combined artificial metrics and deep learning-based functions to build a hybrid model learned by Convolutional Neural Network (CNN). Lin et al. [43] the LSTM model network is used for weak function discovery to find out the cross-item transfer representation of the AST code.

LSTM solved the vanishing gradient problem by inventing the constant error carousel (CEC) unit. The main version of the LSTM block includes unit, input and output gates [44]. The recurrent neural network using LSTM units is usually in a controlled method. On a set of guidance sequences, it is proficient in optimization algorithms (such as gradient descent) and back-propagates over time to calculate the gradient required in the optimization process. The error of the LSTM model network that changes each weight is proportional to the error derivative of the equivalent weight (in the yield layer of the LSTM model network). For ordinary RNNs that use gradient descent, one problem is that between important events with a delay magnitude, the error gradient will quickly disappear. This is because if the spectral radius is less than 1, then $\lim_{n \to \infty} W^n = 0$. However, for LSTM units, when the error value propagates back from the output layer, the error will remain in the cells of the LSTM unit. This "error conveyor belt" continuously feeds errors back to each door of the LSTM unit until they learn to chop off valuable [45].

## Conclusions

SLDeep is learning model [1] has defined the matrix and can improve and optimize, and it also needs the attention layer to save the information. Our work is based on SLDeep that have used only LSTM as the learning model, but our novel model combines LSTM and GRU with the attention mechanism. The technology mentioned above used the measurement of semantic and syntactic information without source code, and the model was suitable for internal or cross-project defect prediction but is used to collect semantic and syntactic information in our proposed AST-based model. Besides, we choose attention mechanisms for crucial features to save information.

The above SDP cannot accurately predict failures in program granular components (such as file level, class level or function level). To solve this problem, we propose a novel framework based on statement-level granularity for high-precision defect prediction DP-AGL.

## Methodology of DP-AGL

To develop an effective SDP structure, we must be inclined to suggest calculations at each sentence level. The whole framework shows the high-level architecture of DP-AGL in Fig 1. The process is divided into eight stages. For the purpose of defining the process of architecture, we

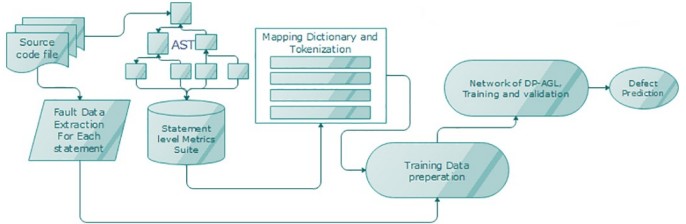

**Fig 1. The high-level overview of the DP-AGL architecture.**

must always outline the relevant metrics of the code to approximate each slanted defect statement. We tend to give away the files of the nodes presented by parsing the source code in Section 3.1. Then, we marked each code statement in section 3.2. We tend to use coaching knowledge to create regional units that are different from the shared knowledge created in the literature. Therefore, we must always use appropriate learning techniques. We tend to suggest a combination of Bi-GRU between Bi-LSTM and LSTM. The characteristics of this technical area unit are carefully applied in Section 3.3. To obtain critical features, we used the attention mechanism in Section 3.4. The new method of our SDP method, especially DP-AGL, is designed by learning sentence slanted sentences through collective actions measured by nodes. We tend to define the subsequent DP-AGL in Section 3.4.

## Parsing source code into nodes

AST (Abstract Syntax Tree) is a suitable picture that reflects the structure and semantic code information. We use Clang 5.0.1 https://clang.llvm.org/index.html tools to develop AST theme programs. Then, we tend to use AST to encrypt the metrics of each subject program. To run our random forest and DP-AGL, we tend to use keras, Tensorflow and sci-kit learning. We are using YACC and LEX tools, tokenization of each statement.

We are familiar with the metric level of 32 node statement to capture complexity. We have defined 22 internal linear metrics and ten external linear metrics for the nodes shown in the Table 1. External linearity measures can capture the features of external discourse that will affect the quality of statements. The internal linear metric estimates the quality of the sentence

**Table 1. The selected metric nodes of ASTs.**

| Internal Nodes | | External Nodes |
| --- | --- | --- |
| Literal String | Pointer count | Function |
| Integer Literal | User-defined function count | Recursive Function |
| Literal count | Function call count | Blocks Count |
| Variable count | Binary operator | Recursive Blocks Count |
| IF Statement | Unary operator | For Block |
| FOR Statement | Compound assignment count | Do Block |
| WHILE Statement | Operator count | While Block |
| DO Statement | Array usage | IF Block |
| SWITCH statement | | SWITCH Block |
| Condition and loop count | | Condition Count |
| Variable Declaration | | |
| Function Declaration Count | | |
| Variable Declaration statement | | |
| Declaration Count | | |

based on the current attributes of the sentence itself. Note that we occupy every row due to the unit of calculation. As an example of the degree of relevance, in the operation of extremely algorithms, the incidence of recursive functions in a statement is an external linear measure. The difference is that the number of binary operators used in the statement is related to the occurrence of internal linear metrics. To obtain effective metrics, we tend to use the insights in the literature to form coarse-grained indicators. Then, we adjust them to obtain fine-grained relevant metrics.

## Embedding tokens for statement metrics

The index column is divided into two categories. The main classes are dedicated to the 32 metrics introduced in Section 3.1 above, which are installed on each program. The second category of a given column is dedicated to each token used in each row. This class is essential because it provides us with a way to capture our structural code. Category 2 can even break any relationship that leads to the introduction of measurement code utilization in Category 1.

The token is the sequence of many characters containing a vocabulary unit in the source code [1]. Each token is finally stored as a pair (t and v), where the type and value of the token correspond. We use a dictionary to identify tokens of IDENTIFIER type. The dictionary is not the only standard in the entire program but not in all programs. These numbers are private based on the order in which they are accessed, and are used as tokenized values for identifiers. We compute to increase the number of tokens across all lines of programs as max(nTi,j). that's why we show an Eq (1)

$$\text{The total number of columns in metrics} = 32 + max(T_{i,j}) \tag{1}$$

T: total number of tokens for the $j^{th}$ row of given program $p_i$, where i is between 1 and p or equal to 1 or p and j is between 1 and $r_j$ or equal to 1 or $r_j$. p is the number of given programs; $r_j$ is the number of rows of program $p_i$.

The tokens in each statement are then padded with the metrics values. For the rows whose tokens are less than the tokens of max(t), we add enough pairs of (0,0) for solving the imbalance problem. Table 2 shows the overall structure of columns of metrics and tokens. The last column shows whether defective or clean.

The general metric and the token information are considered to differentiate the same sequences of statements of code. For example, consider an order of statements as a = 0; c = 2/a; and the second-order of statements as a = 1; c = 2/a; in which they vary only in an integer literal at the very 1st statement. The metric group together with the other extracted tokens will differentiate these two arrangements.

## Bi-LSTM and Gated Recurrent Unit (GRU)

Conventional RNN splits order data in vectors with static length. Each element in the vector symbolizes a particular instant. The result o(t) is not just have influenced for a particular instant t by the up-to-date given x(t) as well as depends over conducted the collection of

**Table 2. The matrix form for each program constructed.**

| Rows/Columns | Metric1 | . . . | Metric32 | Token1 | . . . | Tokenmax | Class |
|---|---|---|---|---|---|---|---|
| $line_1$ | $m_{1,1}$ | . . . | $m_{1,32}$ | $(t_1, v_1)_1$ | . . . | $(t_{max}, v_{max})_1$ | clean/faulty |
| . . . | . . . | . . . | . . . | . . . | . . . | . . . | . . . |

information from the instant t-1 (i.e., h(t-1)), that can be expressed by Eqs (2) and (3):

$$h(t) = f(X(t) * U + h(t-1) * W + b) \tag{2}$$

$$o(t) = g(h(t) * V + c) \tag{3}$$

Among them, U, W, V, b and c represent the weight and deviation of the network, and the activation functions are f and g. The standard recurrent neural network can only remember short-term sequence information, but cannot transmit long-term sequence information. The long and short-term storage unit is mainly composed of associated input gates, associated output gates, and forgetting gates. To prevent the components of the network from disappearing, passing past data (filtered by the forgotten gate), etc. LSTM feeds it and gets current information from the input gate to the output gate. The detail processing of our DP-AGL learning model algorithm is depicted in the *Algorithm* 1

**Algorithm 1**: DP-AGL model learning algorithm

```
Input: D: Dataset;
C: columns containing the code info;
A: Archive of actual code;
l: List codes files of folder;
Source File F = {f₁, f₂, f₃, ..., fₙ};
S: Scanner for CPP programs;
Lit: Defining literals;
Tok: Defining tokens;
Tok2N: Tokens to integer;
N2Tok: Integers to token;
Node of representation NR = {n₁, n₂, n₃, ..., nᵢ};
WL: Weight for loss, importance to binary;
Output: Final accuracy model M
1 Initialize a list of Matrix, a dictionary tok2N and N2Tok;
2 Initialize learning rate lr = 5e-4;
3 Initialize Results for train and test = [];
4 Initialize Verbose = 0;
5 Initialize k = 10;
6 Initialize size: number of codes in each fold = math.ceil(len(l)/k;
7 Initialize units = 250;
8 for i = 1 → k do
9   start = i * size;
10  end = min(len(l), (i + 1) * size);
11  samples from the entire dataset D as training samples
data_train = l[:start] + l[end:];
12  samples from the entire dataset D as testing samples data_test = l
[start: end];
13  if length of data_train and data_test <= 0 then
14    continue;
15  end
16  for j = 1 → units do
17    Predict = model.predict(d_test);
18    Preparing confusion matrix for different measurement for
accuracy;
19  end
20 end
21 return the classification of Train and Test result accuracy;
```

Moreover, to obtain a vast dependence on the bidirectional instants included in the time t. Bi-directional long and short-term memory (Bi-LSTM) can achieve this durability.

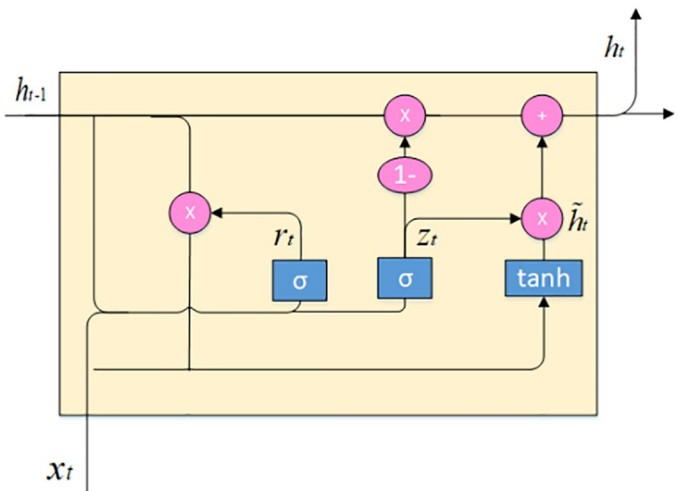

**Fig 2. Gated recurrent unit [12].**

The extension of the GRU model in the given figure is displayed through multiple unified hidden layers. The module structure of GRU is repetitive, which is more straightforward than long and short-term memory because each recurrent neural network feature of the module is the same. It has only two doors, the updated door and the reset door, namely zt and rt in Fig 2. The update gate is used to supervise the extent to which the knowledge of the previously hidden state is extended to the current state. The greater the value of the update gate, the more knowledge of the previous state is introduced. Therefore, if the reset gate is used to adjust the degree of knowledge transfer of the past state, the smaller the value of the reset gate, the more it will be transferred. Therefore, the capture of short-term dependence is usually in the cyclic activation of the reset gate, while the long-term dependence is in the activation of the update gate.

That's why we used Bi-GRU and LSTM with a fully connected layer with an attention mechanism. In our research, the Gated Recurrent Unit (Fig 2) is given by equation Eqs (4) to (8), where $\sigma$ is that the logistic sigmoid function. x and $h_t$ hare the input and therefore the prior hidden state. $W_r$, $W_z$, and Wh are weight matrices that are learned. (The [] indicates that the two vectors are connected, and $^*$ denotes the multiplication of the matrix elements.)

$$r(t) = \sigma(W(r).[h(t-1), x(t)]) \tag{4}$$

$$z(t) = \sigma(W(z).[h(t-1), x(t)]) \tag{5}$$

$$\tilde{h}(t) = \tanh(W\tilde{h}.[r1 * h(t-1), x(t)]) \tag{6}$$

$$h(t) = (1 - z(t)) * h(t-1) + z(t) * \tilde{h}(t) \tag{7}$$

$$y(t) = \sigma(W(o).h(t)) \tag{8}$$

## Attention mechanism

Every time we get the hidden features of the nodes in the sequence from the results of the Bi-LSTM network. These nodes that help explain the meaning of the sequence are not equivalent. To enhance the effect of crucial nodes, after the Bi-LSTM layer, we embed the attention layer. The critical nodes of the meaning of the sequence are clustered together, making the sequence vector necessary for applying the eye contact phenomenon. Its entire process is shown in Fig 3, and we describe it as an Eq (9)

$$u_{it} = \tanh(W_n h_{it} + b_n),$$
$$a_{it} = \frac{\exp(u_{it}^T u_n)}{\sum_t \exp(u_{it}^T u_n)},$$
$$s_i = \sum_t a_{it} h_{it}$$

(9)

That is to say, first mentioned in the Eq (9), we hit the node annotation in the multi-layer perception (MLP) to obtain $u_{it}$. Then, we found the node-level context vector of the hidden symbol of the node $u_n$, which can be checked as the top-level description of the request for the

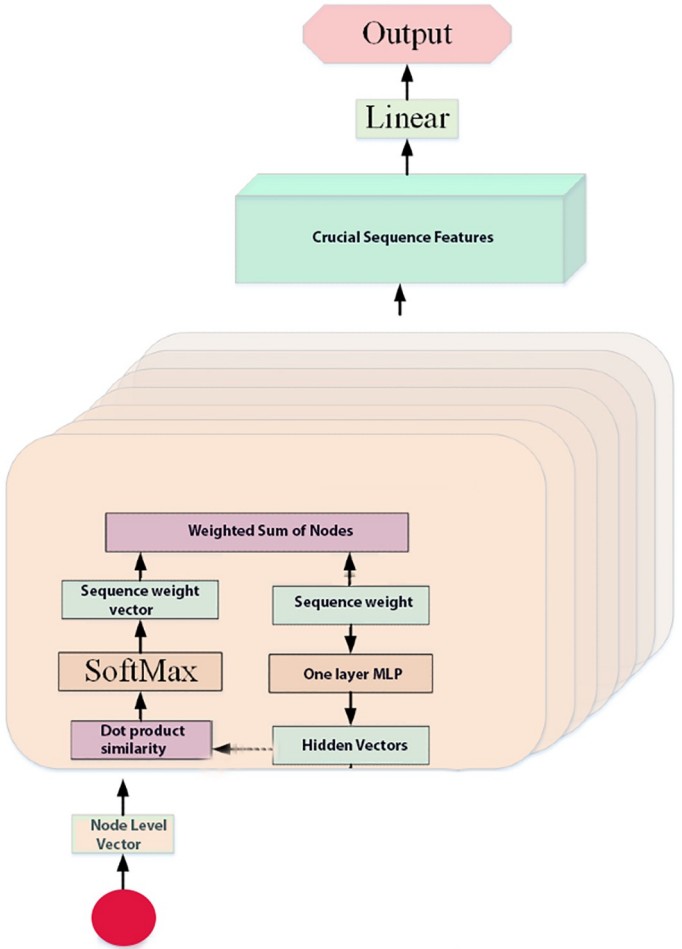

**Fig 3. The process of attention mechanism.**

appearance of the critical node. Then, we check the importance of the node, because the scalar product of $u_{it}$ and $u_n$ is similar, and the importance of the stable $a_{it}$ weight is obtained through the SoftMax function. Finally, as a weighted sum of all nodes, we calculate a sequence of vectors i with related weights. The node-level input of the context vector is randomly initialized and may be updated during the training process.

## Empirical study and analysis

In this segment, we are designing the experiments to authenticate the efficiency of the DP-AGL. Three research questions (RQs) need to be answered as follows:

1. **RQ1**: How DP-AGL method perform with different parameters setting?

2. **RQ2**: Does DP-AGL perform better than basic deep learning techniques, including RF and SLDeep?

3. **RQ3**: Does the DP-AGL method give better performance of fault prediction as compared to state-of-art methods based on static code metrics?

To give answers to the above research questions, we used the DP-AGL architecture shown in Fig 4. In the experiment on the answer to RQ1, we chose TensorFlow and Keras to develop an attention-based two-way-GRU-LSTM network. We also choose dask-jobqueue, dask and dask-ml for machine learning execution on the backend. The implementation of another benchmarking method mainly depends on Python 3.6 and scikit-learn. The experimental operation is set to 2.50 GHz @ 2.71 GHz Intel®Core™i5-7200U CPU, RAM 8.00 GB and 64-bit operating system, based on x64 processor.

The critical components of DP-AGL for learning models are Bi-GRU, Bi-LSTM and attention layer. Our network model contains four layers. The first layer is Bi-LSTM, the second layer is Bi-GRU, the third layer is LSTM, each node has 150 nodes, and the fourth layer is the attention layer. Fig 4 shows the high-level architecture of our proposed network. The first two layers of loops represent feedback. The first layer of LSTM accepts pre-processed data, and vice versa provides a version of the data in the reverse process. The first row of the measurement is read from the first layer to the last row, and then the first row of the measurement is read backwards.

- 150 nodes of Bi-LSTM, with using dropout 0.1 by the recurrent node-set

- 150 nodes of bidirectional GRU, with activation ReLU

- 150 nodes of simple LSTM, with a set of 0.2 dropout

To answer RQ2 to verify the results of Bi-GRU and LSTM on DP-AGL, we supervised different experimental algorithms using different classifiers. The reason why we choose Random Forest (RF) and SLDeep is that the tree-based decision model gives better results in the framework of cross-item [21]. DP-AGL is also a CPDP framework and an application of transfer learning because Benchmark covers various programs and variant sets written by thousands of programmers.

RF is a standard method that includes a collection of classifiers. The purpose of the combination is to assemble the accuracy of the entire model. Each classifier in the random forest method is a decision tree. However, RF proposes a better determination method than a single decision tree.

For the development of predictive models, we tend to run a modified neural network by using the node matrix of the entire program as input. Every program is using indicators. We tend to use the model for our survey information, which involves the use of tenfold cross-

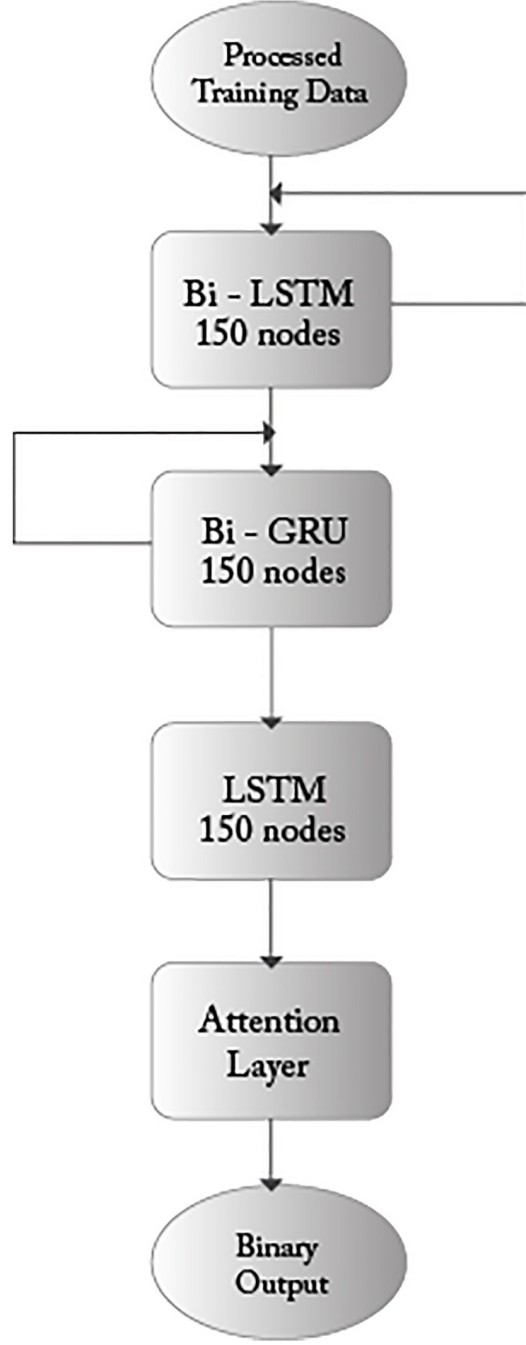

**Fig 4. The high-level topology of DP-AGL learning model.**

validation in a week. Our persistent execution model is due to two motivations. First, consider a lot of training knowledge. Second, for model analysis, we used ten-fold cross-validation. Nevertheless, persistent execution can provide two benefits. First, the subsequent model is used to indicate global attributes because it has learned knowledge from many projects and may be used in many alternative projects.

## Experimental datasets

To answer RQ3, we use Code4Bench for C/C++ code [16], which contains code written by different developers for different problems. Even for every pair of questions and users, there are many versions and defective and clean versions. In the extracted data set, 119,989 subject programs contain 2,356,458 lines of code and 2,920,64 defective lines. For each subject program, the defect data contained in the benchmark. More precisely, for C/C++, the Code4Bench program contains some tables in which the defective version and the correct version are specified.

## Performance evaluation and the results

To evaluate the performance of DP-AGL, we tend to use the software defect prediction literature [20] to calculate four effectiveness metrics, and the area unit is the most important of the same metric. Evaluate area unit accuracy, precision, recall and F-measurement area unit respectively to calculate utilization Eqs (10) to (13). True (TP) represents the number of positive tuples, which contains code lines and code lines that are correctly marked as positive by the classifier in the equation. The true negative number (TN) represents the number of negative tuples that are correctly marked as negative by the classifier. False positives (FP) represent the number of negative tuples containing incorrectly labelled positives. Finally, false negatives (FN) represent the number of positive tuples that contain negatives by correct labelling. In our setup, the defective prone line is positive.

$$Accuracy = \frac{TP + TN}{TP + TN + FP + FN} \tag{10}$$

$$Precision = \frac{TP}{TP + FP} \tag{11}$$

$$Recall = \frac{TP}{TP + FN} \tag{12}$$

$$F - measure = 2 \times \frac{precision \times recall}{precision + recall} \tag{13}$$

The experimental results of DP-AGL are shown in Tables 3 and 4. Table 3 shows the well-trained performance evaluation model, and Table 4 shows the tested performance evaluation model. The corresponding rows of Tables 3 and 4 are shown respectively. Therefore, it means equivalent model training and testing within the same iteration and the same multiple. During ten-fold cross-validation, each slice of the two tables is linked to a specific fold. We have reported the results of applying DP-AGL in the table and applying the Random Forest model and SlDeep model simultaneously according to the subject program. In our experiment, we measured the accuracy of the neighbourhood. For a defective free sentence that is classified as error-prone by the model, if the defective sentence is at most n lines before or after the sentence, we predict it to be true.

In the training and testing phases of Tables 3 and 4 and Figs 5 and 6 with neighbourhoods 4, the average performance indicators have respectively determined DP-AGL, SlDeep and RF. The Tables 5 and 6 shows the average results of Training and Testing with neighbourhoods 2. The chart mainly shows the performance results of DP-AGL, SLDeep and RF. Likewise, the top and bottom charts are similar to the training and testing phases. In the two graphs, the three legends of Precision, Accuracy, and F-measurement are represented by Havelock blue, orange, and dark grey bars, respectively, and the accuracy is displayed in four neighbourhoods.

**Table 3. Experimental results on training with accuracy neighborhood 4.**

| Fold | Model | Recall | Precision | Accuracy | F-measure |
|------|-------|--------|-----------|----------|-----------|
| 1 | DP-AGL | 0.983 | 0.601 | 0.751 | 0.753 |
|   | SLDeep | 0.987 | 0.581 | 0.716 | 0.731 |
|   | RF | 0.20 | 0.578 | 0.630 | 0.30 |
| 2 | DP-AGL | 0.978 | 0.619 | 0.753 | 0.732 |
|   | SLDeep | 0.987 | 0.578 | 0.712 | 0.729 |
|   | RF | 0.195 | 0.582 | 0.629 | 0.292 |
| 3 | DP-AGL | 0.980 | 0.629 | 0.763 | 0.742 |
|   | SLDeep | 0.983 | 0.590 | 0.726 | 0.737 |
|   | RF | 0.179 | 0.576 | 0.627 | 0.273 |
| 4 | DP-AGL | 0.990 | 0.589 | 0.733 | 0.744 |
|   | SLDeep | 0.991 | 0.570 | 0.704 | 0.724 |
|   | RF | 0.180 | 0.578 | 0.615 | 0.275 |
| 5 | DP-AGL | 0.992 | 0.599 | 0.743 | 0.738 |
|   | SLDeep | 0.986 | 0.583 | 0.718 | 0.733 |
|   | RF | 0.144 | 0.576 | 0.623 | 0.231 |
| 6 | DP-AGL | 0.984 | 0.639 | 0.733 | 0.740 |
|   | SLDeep | 0.983 | 0.590 | 0.725 | 0.737 |
|   | RF | 0.174 | 0.575 | 0.624 | 0.267 |
| 7 | DP-AGL | 0.988 | 0.579 | 0.763 | 0.739 |
|   | SLDeep | 0.989 | 0.573 | 0.707 | 0.725 |
|   | RF | 0.205 | 0.574 | 0.628 | 0.302 |
| 8 | DP-AGL | 0.995 | 0.581 | 0.753 | 0.760 |
|   | SLDeep | 0.990 | 0.573 | 0.707 | 0.726 |
|   | RF | 0.194 | 0.574 | 0.627 | 0.290 |
| 9 | DP-AGL | 0.985 | 0.610 | 0.741 | 0.763 |
|   | SLDeep | 0.988 | 0.579 | 0.714 | 0.730 |
|   | RF | 0.201 | 0.574 | 0.627 | 0.297 |
| 10 | DP-AGL | 0.981 | 0.615 | 0.753 | 0.756 |
|   | SLDeep | 0.984 | 0.585 | 0.719 | 0.734 |
|   | RF | 0.13 | 0.578 | 0.620 | 0.216 |
| average | DP-AGL | 0.986 | 0.606 | 0.749 | 0.744 |
|   | SLDeep | 0.987 | 0.580 | 0.715 | 0.731 |
|   | RF | 0.18 | 0.577 | 0.625 | 0.275 |

The graph shows the ten times fold average values. At the peak point legends in graph expressed the DP-AGL model position as compared to others. They are also comparing the each and everything (measurements, models). Both graphs have taken the the some random fold from table of results. The results show how to fit the model on the training data in Table 3. The results show how to fit the model to the test data in Table 4.

## Discussion

We are exploring the weaknesses and strengths of the most advanced models to analyze the DP-AGL learning model. To ensure the validity of the results, we will introduce these aspects in this section.

LSTM is popular because it solves the problem of vanishing gradient. However, when data is moved from one unit to another, problems can arise during evaluation. As the additional

**Table 4. Experimental results on testing with accuracy neighborhood 4.**

| Fold | Model | Recall | Precision | Accuracy | F-measure |
|---|---|---|---|---|---|
| 1 | DP-AGL | 0.978 | 0.579 | 0.703 | 0.712 |
| | SLDeep | 0.989 | 0.554 | 0.675 | 0.708 |
| | RF | 0.187 | 0.550 | 0.612 | 0.279 |
| 2 | DP-AGL | 0.979 | 0.581 | 0.711 | 0.715 |
| | SLDeep | 0.978 | 0.553 | 0.679 | 0.707 |
| | RF | 0.185 | 0.551 | 0.618 | 0.277 |
| 3 | DP-AGL | 0.970 | 0.585 | 0.739 | 0.732 |
| | SLDeep | 0.971 | 0.579 | 0.703 | 0.725 |
| | RF | 0.180 | 0.578 | 0.615 | 0.275 |
| 4 | DP-AGL | 0.971 | 0.583 | 0.716 | 0.713 |
| | SLDeep | 0.985 | 0.554 | 0.681 | 0.709 |
| | RF | 0.179 | 0.580 | 0.627 | 0.274 |
| 5 | DP-AGL | 0.974 | 0.571 | 0.743 | 0.722 |
| | SLDeep | 0.975 | 0.570 | 0.699 | 0.719 |
| | RF | 0.142 | 0.576 | 0.620 | 0.229 |
| 6 | DP-AGL | 0.973 | 0.574 | 0.763 | 0.736 |
| | SLDeep | 0.972 | 0.582 | 0.722 | 0.728 |
| | RF | 0.176 | 0.581 | 0.636 | 0.271 |
| 7 | DP-AGL | 0.980 | 0.578 | 0.773 | 0.742 |
| | SLDeep | 0.981 | 0.588 | 0.721 | 0.735 |
| | RF | 0.220 | 0.608 | 0.637 | 0.323 |
| 8 | DP-AGL | 0.983 | 0.583 | 0.752 | 0.732 |
| | SLDeep | 0.985 | 0.573 | 0.707 | 0.725 |
| | RF | 0.215 | 0.595 | 0.635 | 0.316 |
| 9 | DP-AGL | 0.988 | 0.584 | 0.743 | 0.737 |
| | SLDeep | 0.979 | 0.570 | 0.706 | 0.720 |
| | RF | 0.209 | 0.580 | 0.635 | 0.307 |
| 10 | DP-AGL | 0.983 | 0.598 | 0.754 | 0.736 |
| | SLDeep | 0.977 | 0.576 | 0.719 | 0.725 |
| | RF | 0.12 | 0.585 | 0.635 | 0.202 |
| average | DP-AGL | 0.978 | 0.582 | 0.739 | 0.728 |
| | SLDeep | 0.978 | 0.570 | 0.701 | 0.720 |
| | RF | 0.182 | 0.579 | 0.627 | 0.275 |

function of the door is forgotten, the unit becomes quite complicated. During the LSTM training model, more time is required and a lot of resources are consumed. Since there is a linear layer in each unit, high storage bandwidth is required, and the system usually cannot provide it with high storage bandwidth. Because of dependencies, simple neural networks need to be processed in parallel, so they become better and may provide you with more data. The information in the training model may be lost. Neural networks will slow down over time and experience relative degradation and will not immediately corrode network problems.

The SLDeep model uses a combination of LSTM and 6 neural networks, which increases the complexity of the learning model. In order to solve this problem, we introduced the DP-AGL method that combines Bi-GRU and attention mechanism. The GRU algorithm also uses a gating mechanism to control the memory process. Compared with LSTM, its calculation speed is significantly improved and the complexity is lower. GRU has two gates (reset and update doors). The attention mechanism is to assign high attention weights to the source

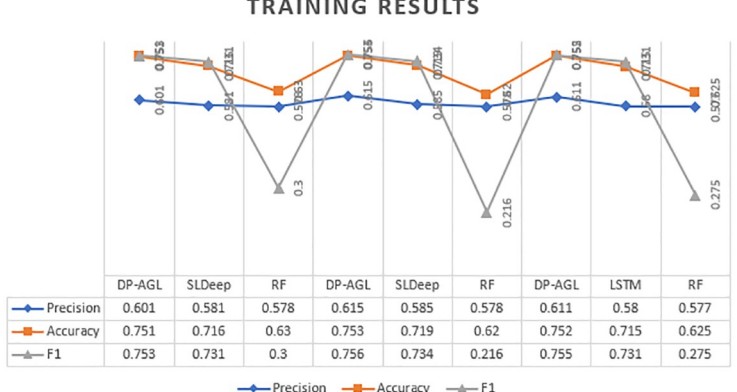

**Fig 5. Training result.**

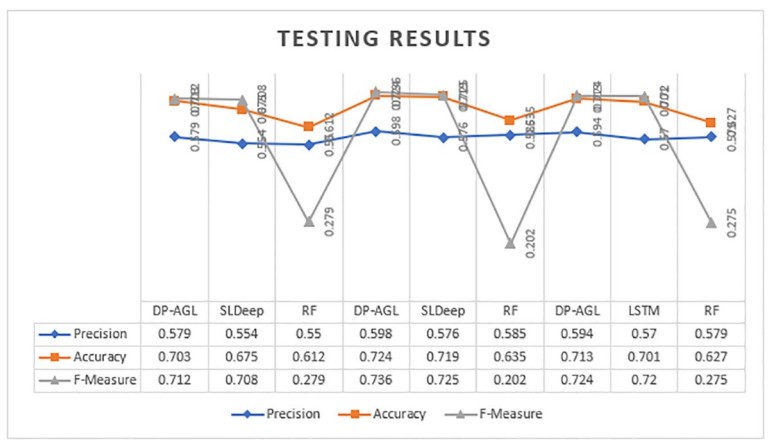

**Fig 6. Testing result.**

**Table 5. Experimental results on training with accuracy neighborhood 2.**

| Fold | Model | Recall | Precision | Accuracy | F-measure |
|------|-------|--------|-----------|----------|-----------|
| average | DP-AGL | 0.981 | 0.511 | 0.659 | 0.624 |
|  | SLDeep | 0.985 | 0.427 | 0.599 | 0.596 |
|  | RF | 0 | 0 | 0.699 | 0 |

**Table 6. Experimental results on testing with accuracy neighborhood 2.**

| Fold | Model | Recall | Precision | Accuracy | F-measure |
|------|-------|--------|-----------|----------|-----------|
| average | DP-AGL | 0.971 | 0.514 | 0.692 | 0.663 |
|  | SLDeep | 0.966 | 0.414 | 0.579 | 0.580 |
|  | RF | 0 | 0 | 0.699 | 0 |

sequences of interest and save them in the missing data after one-to-one attention is paid to them. The source code is available at https://github.com/shahbazshahbaz106/DP-AGL.

## Conclusions

The main purpose of this article is to ensure the safety of the software and reduce the burden on developers by accurately locating fault statements to provide high-quality software with fewer resources and time. In this article, to improve the reliability of the software, we propose a deep learning-based method called DP-AGL (defect prediction through attention-based GRU-LSTM) to support code review and software testing to predict the possible defective code in the software. We defined 32 nodes for the metric and used the learning models Bi-GRU and Bi-LSTM. The attention mechanism is used to extract key features from the output of LSTM. We evaluated DP-AGL on 100,000 Code4Banch C/C++ programs. The average performance of our well-trained DP-AGL model in terms of recall, precision, accuracy and F1 metrics are 0.98, 0.617, 0.75, and 0.757, respectively. DP-AGL is a novel method for sentence-level granularity, which is more effective than SLDeep and Random Forest.

In the future, this research work may be expanded in several ways to improve performance. We can predict defect predictions by using within and between projects. We can design more indicators to extract more features. We can also use these indicators in different languages instead of C/C++ programs. We can choose DP-AGL for Android or commercial or company-based software for evaluation. In the future, this work can be expanded in many ways. We can design more indicators to extract more features. We can also use these indicators in different languages instead of C/C++ programs. We can choose DP-AGL for Android or commercial or company-based software for evaluation.

## Author Contributions

**Conceptualization:** Shengbing Ren, Mubashar Mustafa, Chaudry Naeem Siddique.

**Data curation:** Hafiz Shahbaz Munir.

**Investigation:** Mubashar Mustafa, Chaudry Naeem Siddique.

**Methodology:** Hafiz Shahbaz Munir, Shengbing Ren.

**Project administration:** Shengbing Ren.

**Software:** Hafiz Shahbaz Munir.

**Supervision:** Shengbing Ren.

**Validation:** Hafiz Shahbaz Munir, Shengbing Ren, Mubashar Mustafa, Chaudry Naeem Siddique, Shazib Qayyum.

**Writing – original draft:** Hafiz Shahbaz Munir.

**Writing – review & editing:** Shengbing Ren, Mubashar Mustafa, Chaudry Naeem Siddique, Shazib Qayyum.

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
