## [Decision Letter · Decision Letter 0]

17 Dec 2020

PONE-D-20-33503

Attention Based GRU-LSTM for Software Defect Prediction

PLOS ONE

Dear Dr. Ren,

Thank you for submitting your manuscript to PLOS ONE. After careful consideration, we feel that it has merit but does not fully meet PLOS ONE’s publication criteria as it currently stands. Therefore, we invite you to submit a revised version of the manuscript that addresses the points raised during the review process.

We look forward to receiving your revised manuscript.

Kind regards,

**Le Hoang Son, Ph.D**

Academic Editor

PLOS ONE

**Comments to the Author**

1. Is the manuscript technically sound, and do the data support the conclusions?

Reviewer #1: Yes

Reviewer #2: Yes

2. Has the statistical analysis been performed appropriately and rigorously? 

Reviewer #1: No

Reviewer #2: Yes

3. Have the authors made all data underlying the findings in their manuscript fully available?

Reviewer #1: Yes

Reviewer #2: No

4. Is the manuscript presented in an intelligible fashion and written in standard English?

Reviewer #1: No

Reviewer #2: Yes

5. Review Comments to the Author

**Reviewer #1**:

• In this paper authors propose a deep learning-based method called DP-AGL (defect prediction through attention-based GRU-LSTM) for increasing reliability of software. Although the idea seems to be interesting but significant improvements are required in paper.

• “…internal defect prediction system (WPDP).” Is “WPDP” an abbreviation in section 2.1? If yes then there should be actual words for which this abbreviation is used.

• In section 2.3 the term “SLDeep” is introduced without definition.

• In section 3 “For this reason, we must always outline the relevant metrics of the code to ap-proximate each slanted defect statement.” However, the reason is not mentioned before. Apparently, it seems some disconnect from previous context.

• In section 3, “Then, we marked each code statement in paragraph 3.2.” there is nothing like paragraph 3.2 in paper.

• In equation 1 in section 3.2: “The total number of columns in metrics = 32 + max(nTi;j) (1)” is “nT” a single variable? If yes then use one character as it is confusing in current form.

• In Algorithm 1 DP-AGL model learning algorithm, don’t use serial numbers with inputs. Serial numbers should only be used with executable statements.

• All the equations should be numbered properly.

• First Research question (RQ1) is grammatically incorrect.

• In abstract authors mention about 32 statement level metrics, however, there is no discussion on these metrics in the entire paper.

• Weak experiments and analysis. Compare the proposed model with four to five state of the arts models available in literature. Provide strong discussion with the strengths and weaknesses of the model along with rational.

• Poorly written paper, needs serious revision from an expert. The paper should be cross-checked by any English native speaker.

**Reviewer #2**: 

1. Report all 10-fold experimental result for testing and learning.

2. You only report results by considering radius equal to 4, but the proposed method can act differently with the radius equal to 2

3. In “the three legends of accuracy, accuracy, and F measurement” you wrongly write “accuracy” instead of “precision”

4. Explain more about figure 5 and 6.

5. The only different exists between “SLDeep: Statement-level software defect prediction using deep-learning model on static code features” and your submission is in learning model, and DP-AGL model is based on SLDeep, DP-AGL is too similar to SLDeep. Is there any other novelty in your submission?

6. Authors should upload the code associated to their published article so that readers can view and execute it (e.g. GitHub). Upload your code and link it to your article. It will allow users to re-run the analysis and reproduce the results.

---

## [Author Response · Author response to Decision Letter 0]

30 Dec 2020

Dear Reviewers，

     Thank you for your review of our manuscript (ID: PONE-D-20-33503), which is titled “Attention Based GRU-LSTM for Software Defect Prediction”. We appreciate your concerns and suggestions, and have revised our manuscript accordingly. Please see the attached reviewer response letters.

Sincerely yours,

Hafiz Shahbaz Munir, Shengbing Ren, Mubashar Mustafa, Chaudry Naeem Siddique, Shazib Qayyum

---

## [Decision Letter · Decision Letter 1]

8 Feb 2021

Attention Based GRU-LSTM for Software Defect Prediction

PONE-D-20-33503R1

Dear Dr. Ren,

We’re pleased to inform you that your manuscript has been judged scientifically suitable for publication and will be formally accepted for publication once it meets all outstanding technical requirements.

Kind regards,

**Le Hoang Son, Ph.D**

Academic Editor

PLOS ONE

\\**Comments to the Author**

1. If the authors have adequately addressed your comments raised in a previous round of review and you feel that this manuscript is now acceptable for publication, you may indicate that here to bypass the “Comments to the Author” section, enter your conflict of interest statement in the “Confidential to Editor” section, and submit your "Accept" recommendation.

Reviewer #1: All comments have been addressed

Reviewer #2: All comments have been addressed

2. Is the manuscript technically sound, and do the data support the conclusions?

Reviewer #1: Yes

Reviewer #2: Yes

3. Has the statistical analysis been performed appropriately and rigorously? 

Reviewer #1: Yes

Reviewer #2: Yes

4. Have the authors made all data underlying the findings in their manuscript fully available?

Reviewer #1: Yes

Reviewer #2: Yes

5. Is the manuscript presented in an intelligible fashion and written in standard English?

Reviewer #1: Yes

Reviewer #2: Yes

6. Review Comments to the Author

Reviewer #1: (No Response)

Reviewer #2: (No Response)

---

## [Editor Report · Acceptance letter]

15 Feb 2021

PONE-D-20-33503R1 

Attention Based GRU-LSTM for Software Defect Prediction 

Dear Dr. Ren:

I'm pleased to inform you that your manuscript has been deemed suitable for publication in PLOS ONE. Congratulations! Your manuscript is now with our production department. 

Kind regards, 

on behalf of

Prof. Le Hoang Son 

Academic Editor

PLOS ONE